# Predicting the Exception—CRP and Primary Hip Arthroplasty

**DOI:** 10.3390/jcm10214985

**Published:** 2021-10-27

**Authors:** Marc-Pascal Meier, Ina Juliana Bauer, Arvind K. Maheshwari, Martin Husen, Katharina Jäckle, Jan Hubert, Thelonius Hawellek, Wolfgang Lehmann, Dominik Saul

**Affiliations:** 1Department of Trauma, Orthopedics and Reconstructive Surgery, Georg-August-University of Goettingen, 37075 Göttingen, Germany; marc-pascal.meier@med.uni-goettingen.de (M.-P.M.); inajuliana.bauer@stud.uni-goettingen.de (I.J.B.); katharina.jaeckle@med.uni-goettingen.de (K.J.); Thelonius.hawellek@med.uni-goettingen.de (T.H.); Wolfgang.Lehmann@med.uni-goettingen.de (W.L.); 2Kogod Center on Aging and Division of Endocrinology, Mayo Clinic, Rochester, MN 55905, USA; Maheshwari.Arvind@mayo.edu; 3Department of Orthopedic Surgery, Rochester, MN 55905, USA; Husen.Martin@mayo.edu; 4Department of Trauma, Hand and Reconstructive Surgery, University Hospital Essen, 45147 Essen, Germany; 5Division of Orthopaedics, Department of Trauma and Orthopaedic Surgery, University Medical Center Hamburg-Eppendorf, 20521 Hamburg, Germany; j.hubert@uke.de

**Keywords:** C-reactive protein, CRP, primary hip arthroplasty, orthopedic surgery, revision surgery, periprosthetic joint infection

## Abstract

Background: While primary hip arthroplasty is the most common operative procedure in orthopedic surgery, a periprosthetic joint infection is its most severe complication. Early detection and prediction are crucial. In this study, we aimed to determine the value of postoperative C-reactive protein (CRP) and develop a formula to predict this rare, but devastating complication. Methods: We retrospectively evaluated 708 patients with primary hip arthroplasty. CRP, white blood cell count (WBC), and several patient characteristics were assessed for 20 days following the operative procedure. Results: Eight patients suffered an early acute periprosthetic infection. The maximum CRP predicted an infection with a sensitivity and specificity of 75% and 56.9%, respectively, while a binary logistic regression reached values of 75% and 80%. A multinominal logistic regression, however, was able to predict an early infection with a sensitivity and specificity of 87.5% and 78.9%. With a one-phase decay, 71.6% of the postoperative CRP-variance could be predicted. Conclusion: To predict early acute periprosthetic joint infection after primary hip arthroplasty, a multinominal logistic regression is the most promising approach. Including five parameters, an early infection can be predicted on day 5 after the operative procedure with 87.5% sensitivity, while it can be excluded with 78.9% specificity.


**Highlights**


Seven hundred and eight patients over four years with a primary hip arthroplasty were retrospectively evaluated.On days 11 and 14 post operation, the CRP values were higher in the group with an early infection.As a single parameter, maximum CRP predicts an early infection with 75% sensitivity.A multinominal logistic regression can predict an early infection with a sensitivity and specificity of 87.50% and 78.85%, respectively.This is the first mathematical prediction of early acute periprosthetic infection in primary hip arthroplasty.

## 1. Introduction

Germany is the country with the highest rate of hip replacement surgery within the Organization for Economic Co-operation and Development (OECD). With an incidence of 309 per 100,000 in 2017 and a total number of 243,477 procedures (OPS 5-820) in the year 2019, it ranges in front of Switzerland and Austria [1]. This number is expected to grow further in future decades with the baby boomer generation reaching the age of 65 in 2021 [2,3].

The most relevant complication after primary hip arthroplasty is a periprosthetic joint infection (PJI); these infections also represent the second most frequent postoperative complication in Germany [4,5]. In combination with individual comorbidities and the need for revision surgeries, PJI significantly increases mortality and morbidity [6,7,8]. If a PJI of the hip joint occurs in the first postoperative year, the mortality rate is estimated at around 13.6% [9,10]. In addition to mortality and morbidity, another important aspect of PJIs is the increase in healthcare expenditures. According to United States Nationwide Inpatient Sample (NIS) data, the estimated hospital costs for treating a hip PJI were about USD 30,000 and the average total costs for hip revision surgery due to infection were about USD 80,000 [11,12].

Subsequently, it is crucial to determine a potentially infectious complication early [13]. According to the new PJI-TNM classification published in 2021, early infection is equivalent to stage T0N0Mx. This PJI-TNM classification takes into account implant type and stability, soft tissue conditions, biofilm maturity, pathogen type, patient comorbidities and infection recurrence, overcoming previous classification shortcomings of not capturing PJI diversity [14,15].

To detect an immediate postoperative infection, a thorough clinical wound inspection, regular body temperature measurement and blood tests are generally used. The latter usually include the serum white blood cell count (WBC) and the concentration of C-reactive protein (CRP). The prognostic value of WBC has been demonstrated to be questionable in an aging cohort [16,17], and lack of availability as well as high costs may preclude the broad use of novel markers like interleukin 6 (IL-6) or procalcitonin (PCT).

According to current knowledge, PJIs cannot be detected based on individual CRP values or the WBC [18]. However, to the authors’ knowledge, no study has characterized postoperative CRP kinetics after primary hip arthroplasty and investigated possible correlations with a PJI in the “early stage”. The definition of an early PJI can be traced back to Conventry [19]. It was modified continuously, including by, among others, Tsukayama [20]. However, the time span of a PJI varies greatly between 2 and 12 weeks depending on the definition applied [21]. The definition of an early or late PJI more importantly relies on the degree of maturity of the biofilm. In Germany, common guidelines use a timeframe of 3–4 weeks postoperatively [22,23].

A recent study was able to identify a naturally occurring CRP peak on the third postoperative day after endoprosthetic joint replacement, independently of an infectious complication. The occurrence of a second peak was not considered in detail in this study [24].

One of the major factors contributing to CRP kinetics is the surgical approach. Iorio et al. compared the direct anterior approach (DAA) and direct lateral approach (DLA) and found no correlation between invasiveness (minimally invasive DAA) and CRP levels [25]. In our level-I certified endoprosthesis center in Germany (EPZmax), predominantly anterolateral (ALA) and posterior (PA) approaches are used for primary hip arthroplasty. The present study aimed to record postoperative CRP kinetics after primary hip arthroplasty to define CRP-kinetics in non-infectious and infectious primary hip arthroplasties, and to characterize differences and predictors for an infectious complication. To investigate possible influences of the surgical approach, a comparison between ALA and PA was made.

## 2. Materials and Methods

### 2.1. Patients

We treated 732 patients with primary hip arthroplasty at a level-I certified endoprosthesis center in Germany (EPZmax) between 2016 to 2020. We reviewed these patients retrospectively and included 708 after applying inclusion and exclusion criteria, which are specified below (Appendix A).

The study was approved by the local ethics committee (IRB number 23/4/20, ethics committee of the University of Goettingen) and performed in accordance with the principles expressed in the Declaration of Helsinki. All study participants voluntarily attended the study and gave informed consent.

The collected data included age, sex, length of stay in hospital, day of operation, postoperative time in hospital, diagnosis, indication, type of prosthesis, site of operation, approach, indication, positioning, height, weight, BMI, duration of procedure, infection, revision. Additional serological parameters including CRP, WBC, hemoglobin (Hb) and platelets were recorded.

Serum was collected in the morning, and on average every 2.43 (±0.92) days. The sample collection rate did not differ between the non-infection and infection group (Appendix A). We did not impute CRP values. If revision surgery occurred, we stopped recording serum parameters on the revision day to not confound measurements. Otherwise, the final day of discharge reflected the last day of recorded CRP values.

For all patients, intravenous antibiotics were applied at induction (single shot cefazolin 2 g or 600 mg clindamycin in case of penicillin allergy or vancomycin 1.6 g). According to the hospital’s standards, intraoperatively applied drains were removed on the second day. Wound inspection took place every other day.

All patients were divided into a “non-infection” and “infection” (acute early) group by the authors MPM, IJB and DS. In order to classify a patient to the “infection” cohort, the detection of an organism on a specimen during revision surgery or obvious signs of a purulent infection during revision surgery was necessary. Patients were also assigned into the “infection” cohort by a positive joint puncture result as described below. The indication for revision surgery was given by a standardized sterile joint puncture according to criteria by Izakovicova, Borens and Trampuz [26]. A PJI was defined as a cell count above 2000 or a proportion of more than 70 percent neutrophil granulocytes in the joint puncture [26]. Multiple aerobic and anaerobic tissue biopsies were cultured (at least five with incubations lasting for at least three weeks) during revision surgery. At the same time, histopathology was obtained according to the recommendations of the Centers for Disease Control (CDC) [27,28]. Only the immediate postoperative CRP course was evaluated. After the patient was dismissed from the hospital, no further CRP values were included.

### 2.2. Inclusion Criteria

All patients older than 18 years with a primary hip replacement were included. The selection of patients followed the OPS codes 5-820.0, 5-820.2**, 5-820.4** or 5-820.9 (** means a sixth digit is needed for particular situations). The timeframe was from 1 January 2016 until 31 December 2020.

### 2.3. Exclusion Criteria

Patients with a polytrauma or other surgical interventions were excluded. A preoperatively diagnosed hip tumor or previous hip osteosynthesis (resulting in a Girdlestone situation) were likewise excluded (Appendix A). Every excluded revision surgery was either from a primary operation in another clinic or implanted before the beginning of our study (1 January 2016).

### 2.4. Determination of Peak Value

A postoperative peak was defined as a rise in CRP, which was preceded and followed by lower CRP values. A multiple linear regression analysis was performed to identify parameters determining the maximum CRP. A binary logistic regression for the binary variable “infection” was performed.

### 2.5. Postoperative Kinetics

In non-infection patients, CRP data were plotted and normalized to the peak CRP. Day zero was set to the peak, which led to an exponential decay of CRP values. An approximation with one-phase and two-phase regression was tested. With a satisfactory approximation, we calculated the estimated relative CRP values for days 4 and 5 (49.67% and 43.95% of maximum CRP, respectively). Subsequently, we extended the threshold by 15% (resulting in 72.30% and 64.67% on days 4 and 5) [29,30], and were able to determine values as “failure to decline” for days 4 and 5. Resulting sensitivity and specificity were determined.

### 2.6. Determination of Variables for the Events “Peak CRP” and “Infection”

We aimed to define dependent variables for “maximum CRP”, and performed a backward multiple linear regression (dependent variable: “maximum CRP”). For the single dichotomous variable “infection”, a backward conditional binary logistic regression (dependent variable “infection”) was performed. The combination of variables was assessed in a multinominal logistic regression, as referred to in detail in the results.

### 2.7. Statistics

We a priori estimated the necessary sample size (*t*-test, point biserial correlation model) for a power of 0.95 (estimated effect size ρ = 0.15, α = 0.05). The total necessary sample size was calculated to be *n* = 472 (G*Power 3.1.9.7, Kiel, Germany). We tested for normal distribution of continuous variables with the Anderson–Darling test. Continuous variables were tested with Student’s two-tailed *t*-test and the Mann–Whitney test. Categorial variables were analyzed with the Chi-square test and Fisher’s exact test. Group differences in CRP kinetics were assessed with multiple *t*-tests, adjusted for multiple testing by the Holm–Sidak method.

Overall, mean ± standard deviation was calculated. Statistical analysis was performed with GraphPad Prism 9.00 (GraphPad Software, San Diego, CA, USA), SPSS Statistics software version 27.0 (IBM SPSS Inc., Chicago, IL, USA) and R 4.0.5 (The R Foundation for Statistical Computing, Vienna, Austria). For the PCA analysis, the package PCAtools (2.4.0) was used. For the graphical summary of CRP kinetics, the packages ggplot2 (3.3.5), reshape2 (1.4.4) and tidyverse (1.3.1) were used. Significant differences are marked with asterisks (**** *p* ˂ 0.0001, *** *p* < 0.001, ** *p* < 0.01, * *p* < 0.05).

## 3. Results

### 3.1. Cohort Characteristics: Non-Infection vs. Infection

The operatively treated cohort consisted of 700 patients with no infection, and eight patients with an infection after primary hip arthroplasty. The gender distribution showed a slight imbalance with a predominantly male sex in the infection group (Fisher’s exact, *p* = 0.026, Table 1). Average postoperative days in hospital were significantly longer in the infection group (on average +15.56 days, Mann–Whitney, *p* = 0.044, Table 1). The maximal postoperatively measured CRP was higher in the infection group (Mann–Whitney, *p* = 0.046), while the mean postoperative CRP did not yield differences. Interestingly, the overall maximal CRP (pre- and postoperatively) appeared to be higher in the infection group (Mann–Whitney, *p* = 0.049, Table 1). The existence of a second peak was more frequent in the infection group compared to the non-infection group (Fisher’s exact, *p* = 0.013, Table 1).

### 3.2. Cohort Characteristics: Anterolateral vs. Posterior Approach

While the patients operated on via a posterior approach were younger (Mann–Whitney, *p* ˂ 0.0001, Table 2), and male (Chi-square *p* = 0.026, Table 2), their BMI was higher (Mann–Whitney, *p* ˂ 0.0001) and their preoperative time in hospital was longer (Mann–Whitney, *p* ˂ 0.0001, Table 2). Patients operated on via a posterior approach were hospitalized 1.66 days longer (Mann–Whitney, *p* = 0.002, Table 2), which was in part due to a prolonged postoperative stay (Mann–Whitney, *p* = 0.022, Table 2). The maximal postoperative CRP and overall CRP and the pre- and postoperatively measured WBC, were lower (CRP max and overall Mann–Whitney, *p* = 0.009, respectively, WBC pre- Mann–Whitney, *p* = 0.001 and WBC postoperatively 0.002, Table 2), as well as the overall WBC (Mann–Whitney, *p* ˂ 0.0001, Table 2).

### 3.3. Prediction of Postoperative CRP Kinetics

The comparative kinetics of the pre- and postoperative CRP course in non-infectious and infectious patients is depicted in Figure 1A. On days 11 and 14, the infection group showed significantly higher CRP values compared to the non-infection group (p_adj_ = 0.004, and p_adj_ = 0.002, respectively, two-stage step-up of Benjamini, Krieger and Yekutieli with FDR correction, Figure 1A), while the two approaches (anterolateral vs. posterior) did not differ significantly at any point in time (Figure 1B). Comparing the indications (trauma vs. non-trauma), significant differences occurred on day −1 and day 1 (p_adj_ ≤ 0.0001, respectively, two-stage step-up of Benjamini, Krieger and Yekutieli with FDR correction, Figure 1C).

In order to predict the (postoperative) regularly expected decline of CRP, we assumed a one-phase decay after a peak CRP value was reached. Setting the (individually determined) maximum CRP as day zero and 100%, the predicted one-phase decay is reached in the non-infection group and infection group with a satisfying accuracy (R^2^ = 0.72 and R^2^ = 0.83, respectively). The calculated formula for the one-phase decay was:$$f\left(x\right)=0.7235\times e^{-0.2883\times x}+0.2683$$
for *f*(*x*) = % of CRP peak and *x* = days after maximum CRP (Figure 1D).

### 3.4. Prediction of an Infection: Single Variables

At first, we aimed to predict a PJI by using all assessed parameters in an unbiased principal component analysis (PCA). With all these variables in (unweighted) combination, just an unsatisfying differentiation could be made (Appendix A), so we proceeded with the stepwise validation of single parameters.

#### 3.4.1. Maximum CRP

In order to identify variables most predictive of an infectious complication, we first identified parameters for the maximum CRP, since the maximum CRP was able to differentiate among both the non-infection vs. infection and anterolateral vs. posterior groups. We characterized its predictors performing a multiple linear regression (dependent variable “maximum CRP”, Table 3). In a satisfactory approximation, we constituted six significant parameters determining the maximum CRP. Out of these, the highest weight was attributed to the preoperative mean CRP (β = 0.272, Table 3), followed by the occurrence of a second peak (β = 0.216, Table 3) and the age of the patient (β = 0.161, Table 3). However, applying the maximum CRP value for detection of an infection yielded a maximum sensitivity of 75% and a specificity of 56.86%, respectively (AUC: 0.7028, Figure 2A,B).

#### 3.4.2. Second Peak, Failure to Decline

Next, we assessed the appearance of a second peak (“rise in CRP, which was preceded and followed by lower CRP values”), which was highly predictive in acetabular surgery [29] and spine surgery [30]. The sensitivity and specificity of a second peak were at a low 12.50% and 87.57%, respectively.

A failure to decline could be calculated for days 4 and 5 (for details see methods). The sensitivity and specificity of a “failure to decline” on day 4 were 66.67 and 69.12%, respectively, and for day 5, 50% and 63.50%, respectively. Taken together, the use of a single variable to predict an infection did not lead to satisfactory predictive parameters.

### 3.5. Prediction of an Infection: Binary Logistic Regression

Subsequently, we combined multiple parameters. To determine the probability of an infectious complication, we performed a binary logistic regression with the dependent variable “infection” (backward conditional), and created a model with mediocre approximation (Cox and Snell R^2^ = 0.020, Nagelkerke R^2^ = 0.218, Table 4). In the model, just two out of the six included parameters showed a significant predictive value, preoperative mean CRP (*p* = 0.035, Table 4) and maximum CRP (day, *p* = 0.008, Table 4). The formula was as follows:
$$f\left(x\right)=-2.447*2\mathrm{nd}\;\mathrm{peak}+0.004\times {\mathrm{CRP}}_{\max}+0.029\times \mathrm{BMI}-1.504\times \mathrm{gender}\;+0.381\times {\mathrm{CRP}}_{\max,\mathrm{day}}+0.024\times {\mathrm{CRP}}_{\mathrm{preop}\;\left(\mathrm{average}\right)}-5.85$$

For 2nd peak (1 = yes, 0 = no), gender (1 = male, 2 = female).

This leads to a dynamic sensitivity and specificity model, with an exemplary sensitivity of 75% and specificity of 80.03%, if the cut-off ˃−4.725 is chosen, and an AUC of 0.8297 (Figure 2A,C).

### 3.6. Prediction of an Infection: Multinominal Logistic Regression

In order to improve the mathematical prediction of an emerging infection, we performed a multinominal logistic regression with five variables: approach (1–6, ordinal), preoperative mean CRP (numerical), day of maximum CRP (numerical), gender (dichotomous), and failure to decline on day 5 (dichotomous), leading to a dynamic sensitivity and specificity, depending on a specific cut-off. The obtained formula was:$$f\left(x\right)=1.957\times \mathrm{approach}+0.0041\times {\mathrm{CRP}}_{\mathrm{preop}\;\mathrm{mean}}+0.611\times {\mathrm{CRP}}_{\max,\mathrm{day}}\;-18.57\times \mathrm{gender}+1.965\times {\mathrm{failure}\;\mathrm{to}\;\mathrm{decline}}_{\mathrm{day}\;5}+4.988$$

For approach [1 = anterolateral (Watson-Jones), 2 = posterior, 3 = lateral (Bauer), 4 = anterior intrapelvic (STOPPA), 5 = Kocher-Langenbeck, 6 = anterior], gender (1 = male, 2 = female), failure to decline_day5_ (1 = yes, 0 = no).

We depict the resulting ROC analysis with an area under the curve (AUC) of 0.8757 in Figure 2A,D. Exemplarily, setting a cut-off ˃−8.566 leads to a sensitivity of 87.5% to detect an infectious complication and an accompanying specificity of 78.85% to not miss an infection (Figure 2D).

### 3.7. Patients with Infections

From the eight patients with a postoperative infection, four (50%) suffered from an *S. aureus* infection. In just one patient, a second peak occurred, while two (25%) showed a failure to decline (Table 5).

## 4. Discussion

The development of a PJI prolongs the inpatient stay considerably [31,32]. The maximum CRP and the presence of a second peak can be helpful for early infection detection [33,34,35]. This study was able to demonstrate the clinical applicability of two formulas forecasting a PJI. In addition, it was noticeable that the patients operated on via the posterior approach had a slightly longer length-of-stay, lower age and higher BMI than those operated on via the anterolateral approach.

Comparing patients without and with a postoperative PJI, we observed a longer postoperative stay in hospital in the patients with infection. On average, the patients with an acute early PJI stayed about 15.5 days longer than the patients without an infection. Data in the US shows an increase in the annual treatment costs of a PJI from 320 to 566 million USD/year from 2001 to 2009, which is projected to rise to USD 1.62 billion per year [12]. In Denmark, the cost per patient of a septic revision is EUR 27,059, compared to an aseptic revision at EUR 14,760 [36], with these differences illustrating how a complicating PJI after primary hip arthroplasty has an enormous economic impact.

An important parameter for the postoperative assessment of an infection is serum CRP values [37,38]. Contrary to the study by Akgün et al. [39], the present study was able to demonstrate the utility of individual CRP values for the detection of a PJI. We demonstrate differences between the patients without and with a PJI for the postoperative maximum CRP as well as for the overall CRP. Comparable results were found by Praz et al. This group analyzed serum CRP kinetics and found significantly higher serum CRP values in septic revision after primary hip and knee arthroplasties. The authors describe a sensitivity and specificity of 87.5% and 86.1%, respectively, in detecting a PJI with serum CRP values (cut-off 9 mg/L) [40]. When we used just a single value (maximum CRP), we were able to achieve a comparable sensitivity of 75%, but lower specificity of 57%. An explanation for the difference might have been that our PJI group was relatively small, while Praz et al. examined 42 revisions caused by a PJI. In addition, Sigmund et al. reported a relatively low sensitivity of 68%, but a higher specificity of 87% for serum CRP values alone to predict a PJI (cut-off 10 mg/L) in 177 patients undergoing hip or knee arthroplasty and revision surgery. In this study, 75 cases were classified as septic revisions, with serum CRP having an AUC of 0.78 [41]. Taking all these studies together, it should be noted that a maximum serum CRP value alone can be used to detect a PJI after hip arthroplasty. However, serum CRP values alone should not be used to detect a periprosthetic joint infection after hip arthroplasty, as mentioned by several guidelines and large patient cohorts [39,42,43,44,45,46]. A recent study on 177 patients confirmed the accuracy of serum CRP values in detecting a PJI, but concluded that CRP values alone can just be a suggestive criterion, while it should be complemented by more specific tests (i.e., synovial analysis) [41], which has also been found by another retrospective study with 215 patients [39].

Similarly, our study was able to demonstrate that a postoperative infection after treatment of an acetabular fracture can be detected by the maximum CRP and a second peak with a sensitivity of 83% and a specificity of 81% [29].

The previous remarks illustrate the importance of individual CRP values in the detection of a PJI. Another important aspect in laboratory diagnostics is CRP kinetics [38]. In order to use the dynamics of CRP development, we transformed the CRP values mathematically to investigate CRP kinetics in a relative manner from their maximum value. While the patients without an infection rarely revealed a second peak, it was more frequent in the patients with infection. The combination of a second peak and failure to decline seemed to be highly predictive in acetabular and spine surgery [29,30], while in this study, we could not confirm these results in primary hip arthroplasties. This may be due to the low numbers (four) in our cohort having a second CRP peak, out of eight patients with a PJI. Another reason might have been the traumatic kind of acetabular and vertebral fractures, which were less frequent in our analysis. Trauma itself is known to affect changes in CRP kinetics [47]. More than 60% of the indications for surgical treatment in our collective were of a nontraumatic origin.

To further elucidate the dynamic CRP changes, we made use of a logistic regression, which resulted in a sensitivity and specificity of 75% and 80.03%, respectively (cut-off ˃−4.725, AUC of 0.8297) for predicting a hip PJI. To further improve the sensitivity, we utilized a multinomial logistic regression and achieved a sensitivity and specificity of 87.5% and 78.85%, respectively (AUC: 0.8757). Potentially due to the higher complexity of the formula used, our analysis yielded significantly better results than comparative studies. Erdemli et al. achieved an AUC of 0.644 in a similar study [48]. They examined 88 patients who underwent revision arthroplasty, and compared a PJI-group (*n* = 36) to an aseptic-group (*n* = 52). In comparison to our study the authors examined a higher number of patients with infection. Schutz et al. [49] achieved a sensitivity of 65% and a specificity of 85% with two CRP values above their correlation, which was calculated with the help of a linear mixed model. Their 42 patients included hip arthroplasties as well as osteosynthetic treatments. The inclusion of osteosyntheses could have possibly influenced their analysis, while their overall number of patients was lower than ours [49]. Klim et al. similarly investigated the predictive value of CRP kinetics in the detection of a PJI in total hip or knee arthroplasties. Based on their ROC analysis, they concluded that a combined serum biomarker analysis had no benefit in the early diagnosis of a PJI [50]. Klim et al. examined 84 patients after total joint replacement of the knee or hip, of which a PJI was diagnosed in 55 cases. These authors examined significantly more PJIs than we did in our study. The contrary statement to our study could be due to the statistical methods or procedures of the authors. In contrast to our multinomial logistic regression, they performed a logistic regression with lasso regularization, and included knee arthroplasties in their study of 124 patients [50]. Schinsky et al. established that better predictive values can be achieved with the inclusion of a joint puncture in the synopsis of WBC in the aspirate, an elevated erythrocyte sedimentation rate (ESR) and CRP levels [51]. Taking their results into account, the predictive power of our formula could possibly be improved in the future by an additional joint puncture.

The question of “the optimal” approach for implantation of a primary hip joint endoprosthesis is still controversially discussed throughout the literature [52,53,54,55,56,57,58,59]. In our clinic, the anterolateral (Watson-Jones) and posterior (Moore) approaches [60,61] were predominantly used. Anatomical advantages and disadvantages of the two surgical approaches have been discussed extensively elsewhere [58,59].

Of note, the patients who were operated on via the posterior approach stayed in our hospital slightly longer. This is in agreement with a study by Wang et al., showing a significantly shorter length of stay for the anterolateral approach (6.4 ± 2.2 days) compared to the posterior approach (9.2 ± 3.1 days) in a study with 42 patients with an average age of 78.1 years [62].

We found no significant differences between the two approaches regarding the occurrence of a PJI. When comparing surgical approaches, recent studies were similarly unable to demonstrate a clear access-related disposition to infection. Shohat et al. showed that the direct anterior approach (DAA) to the hip does not increase the risk for a subsequent PJI [63]. No difference of DAA and direct lateral approach (DLA) regarding CRP values on days 1, 3 and 5 was found by Iorio et al., while the operative time was significantly shorter in DLA [25]. The surgeons included in this study used their individual favored approach, which may explain the lack of difference seen between the two approaches. This is comparable to our study, in which all surgeons used their individually preferred technique. Several studies have already shown that the access-related operating time is highly dependent upon the surgeon’s experience [59]. Maezawa et al. reported a difference in CRP levels between the direct anterior approach (DAA) and posterior approach (PA) on day 1 after the operation, but not on day 4 in a study comprising 71 women [64]. As the number of overall cases in this study is rather small compared to our study and the influence of possible previous diseases is not considered, the reported significant difference on the first postoperative day could be a detection bias.

## 5. Limitations

By its nature, the low number of infections in primary hip arthroplasty restricts generalization of the results. We cannot exclude the possibility of infectious complications after the dismissal of patients; however, this seems unlikely due to the setting of the clinic and severity of a potential PJI. Nevertheless, some less virulent microorganisms may have led to the delayed development of symptoms and a delayed or late infection may have been missed, lowering the generalization of our results.

We did not assess further laboratory parameters like PCT and IL-6 on a regular basis, and therefore this has not been included in the present investigation. In future studies, these will be compared with CRP values and other traditional methods of infection detection prospectively. In addition, the results of this study will be tested prospectively in a different patient cohort.

## 6. Conclusions

The CRP kinetics after regular primary hip arthroplasty can be predicted after reaching the maximum CRP with an R^2^ of 0.7159. The peak CRP is directly affected by the preoperative mean CRP, the appearance of a second peak, age, and gender. An acute infection can be accurately predicted with a binary logistic regression resulting in a sensitivity of 75% and a specificity of 80%. This is an indication of a reliable dynamic CRP development, which should be more focused on multiple than single values in the evaluation of a PJI. Single CRP values should not be used to diagnose a PJI or for the decision to perform a revision, as recommended by several guidelines, but complemented by more specific tests [44,45,46].

A more complex multinominal logistic regression leads to a sensitivity and specificity of 87.50% and 78.85%, respectively.

Summarizing, a one-phase exponential decay can predict CRP kinetics, and an acute infectious complication can be reliably projected by using just five parameters. This easily applicable and budget-friendly formula represents a useful additional tool to further guide physicians to prognosticate the need for hip arthroplasty revision.

## Figures and Tables

**Figure 1 jcm-10-04985-f001:**
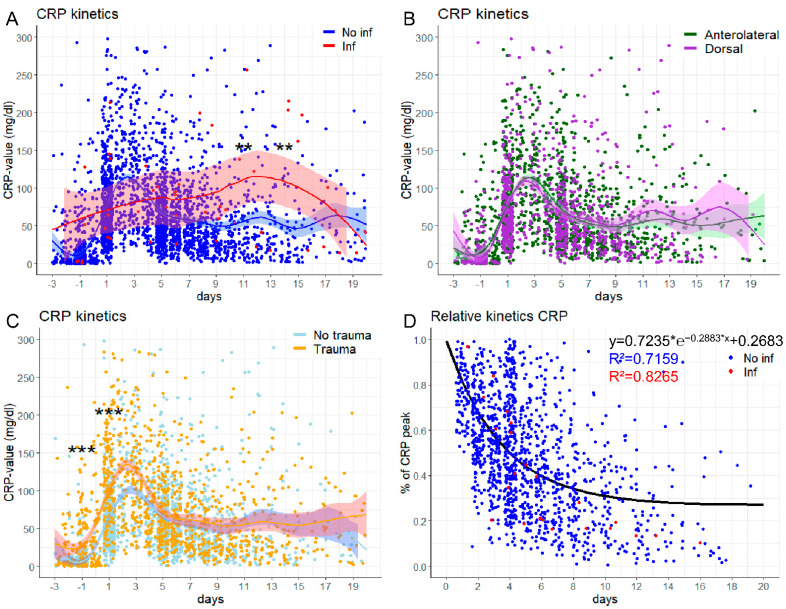
The postoperative CRP course in patients with and without an infection. (**A**) Comparing the absolute CRP course in patients with (red) and without (blue) an infection demonstrates a significant difference on days 11 and 14 (FDR day 11: q-value = 0.0037, FDR day 14: q-value = 0.0019). (**B**) Comparing the anterolateral (green) with the posterior (orchid) approach, no statistically significant differences regarding daily CRP can be detected. (**C**) Comparing traumatic and nontraumatic hip arthroplasty, traumatic have significantly higher CRP values on days −1 (FDR q-value ˂ 0.0001) and 1 (FDR q-value < 0.0001). (**D**) The relative CRP kinetics after a maximum CRP value follows a one-phase exponential decay and can be predicted in a sufficient manner (R^2^ no infection: 0.716, R^2^ with an infection: 0.827). *** *p* < 0.001, ** *p* < 0.01. * means multiplication.

**Figure 2 jcm-10-04985-f002:**
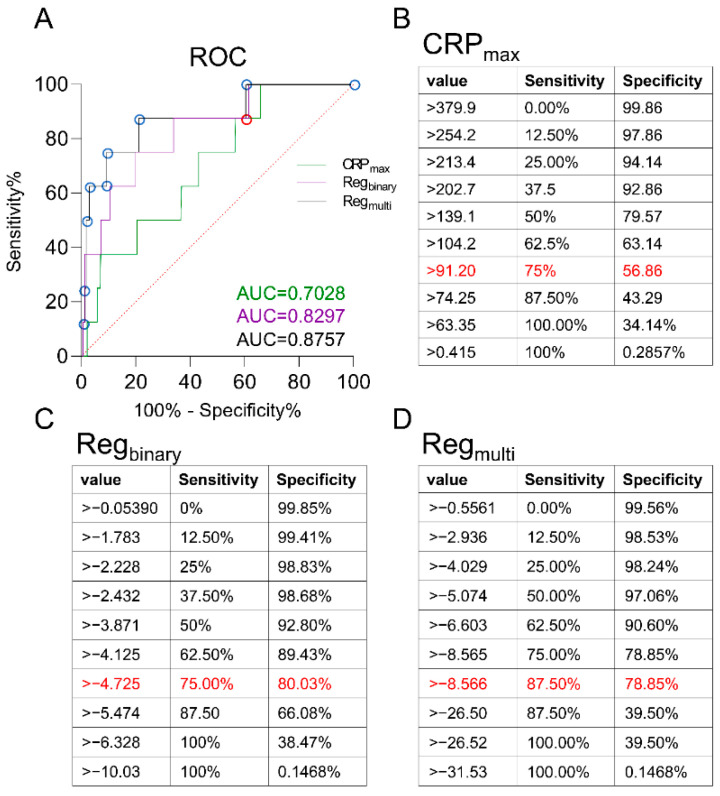
ROC analysis of a single value and two formulas for the prediction of an infection. (**A**) The ROC analysis shows the different performance of three infection prediction models. (**B**) The single parameter “CRP max” is able to predict an infection with a sensitivity of 75% and a specificity of 56.86%, if the cut-off is set above 91.20 mg/L (AUC: 0.7028). (**C**) A binary logistic regression with a cut-off above −4.725 leads to a sensitivity and specificity of 75% and 80.03%, respectively (AUC: 0.8297). (**D**) A multinominal logistic regression consisting of the five most important parameters leads to the development of an equation with the ability to predict an infectious complication with a sensitivity of 87.5% and an accompanying specificity of 78.85%, if the cut-off is set to >−8.566 (AUC: 0.8757). The mentioned “best” values are marked in red.

**Table 1 jcm-10-04985-t001:** Group characteristics in the non-infection and infection cohort.

Parameter		Non-Infection (±SD) (*n* = 700)	Infection (±SD) (*n* = 8)	*p*-Value
Age (years)		71.18 (±11.92)	68.38 (±6.02)	0.325 (n.s.) ^1^
Sex				**0.026** (*) ^2^
	Male	44.86%	87.5%	
	Female	55.14%	12.5%	
BMI (kg/m^2^)		27.08 (±5.40)	27.57 (±4.29)	0.382 (n.s.) ^1^
Days until surgery		1.66 (±3.19)	3.38 (±5.01)	0.237 (n.s.) ^1^
Days in hospital (total)		11.94 (±5.67)	29.25 (±22.89)	0.053 (n.s.) ^1^
Days in hospital (postop)		10.30 (±4.25)	25.86 (±20.95)	**0.044** (*) ^1^
Time operation (min)		106.04 (±37.65)	117.8 (±53.96)	0.655 (n.s.) ^1^
Preoperative CRP (mean)		13.07 (±25.90)	23.24 (±41.99)	0.806 (n.s.) ^1^
Preoperative CRP (max)		14.17 (±28.30)	29.49 (±58.57)	0.821 (n.s.) ^1^
Postoperative CRP (max)		97.60 (±60.95)	143.38 (±72.31)	**0.046** (*) ^1^
Postoperative CRP (mean)		65.89 (±37.57)	82.92 (±42.20)	0.200 (n.s.) ^1^
CRP overall (max)		98.06 (±60.99)	143.38(±72.31)	**0.049** (*) ^1^
CRP overall (mean)		52.41 (±33.10)	73.40 (±41.77)	0.109 (n.s.) ^1^
Average day of 2nd CRP peak		9.46 (±3.15)	8.25 (±3.20)	0.516 (n.s.) ^1^
Preoperative WBC (mean)		8.45 (±3.89)	7.59 (±2.55)	0.390 (n.s.) ^1^
Postoperative WBC (max)		10.41 (±5.11)	11.01 (±5.30)	0.829 (n.s.) ^1^
WBC overall (max)		10.93 (±5.54)	11.28 (±5.24)	0.923 (n.s.) ^1^
Preoperative Hb (mean)		13.43 (±2.06)	12.32 (±1.89)	0.105 (n.s.) ^1^
Postoperative Hb (max)		11.07 (±3.00)	10.05 (±0.86)	0.130 (n.s.) ^1^
Hb overall (max)		13.71 (±3.04)	12.4 (±1.71)	0.103 (n.s.) ^1^
Approach				0.151 (n.s.) ^3^
	Anterolateral (Watson-Jones)	50.86%	25%	
	posterior	40.71%	50%	
	Lateral (Bauer)	4.71%	25%	
	Anterior intrapelvic (STOPPA)	0%	0%	
	Kocher-Langenbeck	0.14%	0%	
	anterior	0.29%	0%	
Emergency-Ind?				0.724 (n.s.) ^2^
	Yes	33.00%	37.50%	
	No	67.00%	62.50%	
Indication				1.000 (n.s.) ^2^
	Trauma	34.29%	37.5%	
	Other	65.71%	62.5%	
2nd CRP peak				**0.013** (*) ^2^
	yes	12.86%	50%	
	no	87.14%	50%	

^1^ Mann–Whitney test, ^2^ Fisher’s exact test, ^3^ Chi-square test., * *p* < 0.05, significant differences are highlighted in **bold.**

**Table 2 jcm-10-04985-t002:** Group characteristics in the non-infection group: anterolateral vs. posterior approach.

Parameter		Anterolateral (±SD) (*n* = 358)	Posterior (±SD) (*n* = 289)	*p*-Value
Age (years)		73.74 (±11.69)	67.51 (±11.46)	**<0.0001 (****) ^1^** (n.s.) ^1^
Sex				**0.026 (*) ^2^**
Male		42.46%	49.66%	
Female		57.54%	50.34%	
BMI (kg/m^2^)		25.92 (±4.58)	28.63 (±6.16)	**<0.0001 (****) ^1^**
Days until surgery		1.43 (±2.42)	2.10 (±4.17)	**<0.0001 (****) ^1^**
Days in hospital (total)		11.64 (±5.82)	13.03 (±7.02)	**0.002 (**) ^1^**
Days in hospital (postop)		10.25 (±4.94)	10.93 (±4.98)	**0.022 (*) ^1^**
Time operation (min)		105.69 (±36.43)	108.23 (±41.18)	0.391 (n.s.) ^1^
Preoperative CRP (mean)		13.89 (±25.35)	11.95 (±25.22)	0.371 (n.s.) ^1^
Preoperative CRP (max)		15.30 (±28.28)	13.99 (±28.33)	0.459 (n.s.) ^1^
Postoperative CRP (max)		102.47 (±59.63)	93.90 (±63.91)	**0.009 (**) ^1^**
Postoperative CRP (mean)		67.26 (±35.59)	65.34 (±41.01)	0.090 (n.s.) ^1^
CRP overall (max)		102.77 (±59.47)	94.42 (±64.15)	**0.009 (**) ^1^**
CRP overall (mean)		53.84 (±31.40)	51.68 (±36.18)	0.072 (n.s.) ^1^
Average day of 2nd peak		9.45 (±3.06)	9.39 (±3.37)	0.749 (n.s.) ^1^
Preoperative WBC (mean)		8.88 (±4.70)	7.80 (±2.52)	**0.001 (**) ^1^**
Postoperative WBC (max)		10.55 (±4.10)	10.32 (±6.43)	**0.002 (**) ^1^**
WBC overall (max)		11.18 (±4.99)	10.64 (±6.45)	**<0.0001 (****) ^1^**
Preoperative Hb (mean)		13.43 (±1.89)	13.45 (±2.35)	0.638 (n.s.) ^1^
Postoperative Hb (max)		10.93 (±2.69)	11.29 (±3.44)	0.125 (n.s.) ^1^
Hb overall (max)		13.64 (±2.64)	13.85 (±3.60)	0.910 (n.s.) ^1^
Infection				0.415 (n.s.) ³
Yes		0.56%	1.38%	
No		99.44%	98.62%	
2nd peak				0.389 (n.s.) ^2^
		yes	14.80%	12.46%
		no	85.20%	87.54%

^1^ Mann–Whitney test, ^2^ Chi-Quadrat test ³ Fisher’s exact test. **** *p* ˂ 0.0001, *** *p* < 0.001, ** *p* < 0.01, * *p* < 0.05, significant differences are highlighted in **bold.**

**Table 3 jcm-10-04985-t003:** Multiple linear regression with “maximum CRP” as the dependent variable (“backward”, R^2^ = 0.284).

Independent Variable	Standardized Coefficients β	*p*-Value
CRP (preop mean)	0.272	0.000 (****)
Appearance of a 2nd peak	0.216	0.000 (****)
Age	0.161	0.001 (**)
Postoperative days in hospital	0.126	0.008 (**)
Failure to decline on day 5	−0.125	0.006 (**)
Gender (1 = m)	−0.110	0.015 (*)

**** *p* ˂ 0.0001, ** *p* < 0.01, * *p* < 0.05.

**Table 4 jcm-10-04985-t004:** Binary logistic regression with “infection” as the dependent variable (“backward: conditional”, Cox and Snell R^2^ = 0.020, Nagelkerke R^2^ = 0.218).

Covariate Variable	Exp	*p*-Value
Second peak	0.087	0.186 (n.s.)
CRP max	1.004	0.410 (n.s.)
BMI	1.030	0.764 (n.s.)
Sex	0.222	0.217 (n.s.)
CRP (preop mean)	1.024	**0.035 (*)**
CRP (max day)	1.464	**0.008 (**)**

** *p* < 0.01, * *p* < 0.05.

**Table 5 jcm-10-04985-t005:** Infectious complications occurred in eight patients.

Pat.	Peak (Day)	Peak (mg/L)	Specimen	2nd CRP Peak (Day)	Failure to Decline?	Predicted CRP (mg/L)	Actual CRP (mg/L)
1	11	256.4	*S. aureus*		No	165.81 (day 4)	-
2	6	91.5	*S. aureus*		No	59.17 (day 4)	-
3	6	105.1	*P. aeroginosa*		Yes	67.96 (day 4)	96.60
4	6	63.4	*S. aureus*		No	41.00 (day 4)	-
5	1	139.1	*S. aureus*		No	89.95 (day 4)	28.30
6	5	74.6	*Staphylococcus saccharolyticus*		Yes	53.93 (day 3)	68.9
7	14	203.2	*Enterococcus*	11	No	131.4 (day 4)	-
8	1	213.7	*P. aeroginosa*		No	138.19 (day 4)	129.50

## Data Availability

Data available on request due to restrictions e.g., privacy or ethical.

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
