# Peer review of "Predicting the Exception—CRP and Primary Hip Arthroplasty"

_jcm, 2021, doi:10.3390/jcm10214985_

Round 1
Reviewer 1 Report
Dear Authors,
you aimed to validate the value of postoperative C-reactive protein (CRP) and develop a formula to predict a periprosthetic joint infection, by evaluating 708 patients with primary hip arthroplasty retrospectively. The maximum CRP predicted an infection with a sensitivity and specificity of 75% and 56.9%, respectively, while a binary logistic regression reached values of 75% and 80%. A multinominal logistic regression, however, was able to predict an infection with a sensitivity and specificity of 87.5% and 78.9%.
Although the English language and style is fine and the quality of the presentation is good some minor corrections are required;
-page 3 line 1: I would use “treated with” instead of “treated by”
-the quality of suppl. 1 is not sufficient.
-the layout of the manuscript has to be improved, as some headlines should be placed right at the top of the paragraph to make the manuscript easier to read.
In your conclusion, you mentioned that a multinominal logistic regression is the most promising approach to predict early acute periprosthetic joint infection after primary hip arthroplasty. Another important aspect in your study is the value of CRP kinetics in laboratory diagnostics. It is important to use the dynamics of CRP development for the evaluation in PJI.
Although these results are of interest for the readers and the scientific community, it should be noted that serum CRP values alone should not be used to detect a periprosthetic joint infection after hip arthroplasty, as mentioned in several infection guidelines and in several studies including large patient cohorts.
A recent published study, which evaluated novel serum inflammatory markers in the diagnosis of periprosthetic joint infections, could show that although serum CRP shows the best accuracies among the widely available serum inflammatory parameters, these biomarkers can only be recommended as suggestive criteria in diagnosing PJI. The preoperative workup should always be complemented by more specific tests such as synovial fluid analysis.
I could recommend to publish that paper, if the authors would rewrite their conclusion section and mention that the CRP values should not be used as a single parameter to diagnose an PJI and furthermore should not be used for the decision to perform a revision.
Author Response
Dear Authors,
you aimed to validate the value of postoperative C-reactive protein (CRP) and develop a formula to predict a periprosthetic joint infection, by evaluating 708 patients with primary hip arthroplasty retrospectively. The maximum CRP predicted an infection with a sensitivity and specificity of 75% and 56.9%, respectively, while a binary logistic regression reached values of 75% and 80%. A multinominal logistic regression, however, was able to predict an infection with a sensitivity and specificity of 87.5% and 78.9%.
Although the English language and style is fine and the quality of the presentation is good some minor corrections are required;
We thank the reviewer for the positive feedback and constructive comments on our manuscript.
-page 3 line 1: I would use “treated with” instead of “treated by”
We thank the author for this suggestion and changed the sentence as follows: “We treated 732 patients with primary hip arthroplasty (…)”.
-the quality of suppl. 1 is not sufficient.
We apologize for the lack of figure quality. We originally provided a 600 dpi tif file, however, the quality of this figure in the pdf is indeed insufficient. We upscaled the figure and now provide a 1,200 dpi tif.
-the layout of the manuscript has to be improved, as some headlines should be placed right at the top of the paragraph to make the manuscript easier to read.
We agree with the reviewer and provided a redesigned format of our manuscript. As we can only submit a Word file, we will ask the editorial board to make these adjustments in the pdf creation. In addition, we submitted a revised version of Table 1, since this was disarranged in the supplied pdf.
In your conclusion, you mentioned that a multinominal logistic regression is the most promising approach to predict early acute periprosthetic joint infection after primary hip arthroplasty. Another important aspect in your study is the value of CRP kinetics in laboratory diagnostics. It is important to use the dynamics of CRP development for the evaluation in PJI.
We agree with the reviewer and are grateful for this suggestion. To focus the reader on this, we added following text to the conclusion: “This is an indication of a reliable dynamic CRP development, which should be more focused on than single values in the evaluation of a PJI.”.
Although these results are of interest for the readers and the scientific community, it should be noted that serum CRP values alone should not be used to detect a periprosthetic joint infection after hip arthroplasty, as mentioned in several infection guidelines and in several studies including large patient cohorts.
We thank the reviewer and added “However, serum CRP values alone should not be used to detect a periprosthetic joint infection after hip arthroplasty, as mentioned by several guidelines and large patient cohorts [39,42–46].” to the results section.
A recent published study, which evaluated novel serum inflammatory markers in the diagnosis of periprosthetic joint infections, could show that although serum CRP shows the best accuracies among the widely available serum inflammatory parameters, these biomarkers can only be recommended as suggestive criteria in diagnosing PJI. The preoperative workup should always be complemented by more specific tests such as synovial fluid analysis.
We thank the reviewer for this comment and the very interesting publication suggested. After studying this manuscript, we added following paragraph to the results: “A recent study on 177 patients confirmed the accuracy of serum CRP values in detecting a PJI, but concluded that CRP values alone can just be a suggestive criterion, while it should be complemented by more specific tests (i.e. synovial analysis) [41], which has also been found by another retrospective study with 215 patients [39].”.
I could recommend to publish that paper, if the authors would rewrite their conclusion section and mention that the CRP values should not be used as a single parameter to diagnose an PJI and furthermore should not be used for the decision to perform a revision.
We thank the reviewer for this suggestion and the opportunity to improve our manuscript. We added “Single CRP values should not be used to diagnose a PJI or for the decision to perform a revision, as recommended by several guidelines, but complemented by more specific tests [44–46].” to the conclusion.

Reviewer 2 Report
First of all, I want to congratulate the authors for your work. I think it is very interesting to work on clinical challenges and try to demonstrate that just with CRP kinetics we can anticipate a so difficult diagnosis as it is PJI.
In the abstract you say that you aim to "validate" de value of CRP to predict of PJI. I think what you describe very well are the different approaches to obtain maximal value of CRP kinetics in your series, but to validate the usefulness of your best approach, I think you have to apply in a different cohort of the one that you have used to obtain sensitivity and specificity in your study.
In the inclusion and exclusion criteria I haven´t understood why revision surgeries were excluded. The 60 cases were so precocious that didn´t permit to measure the CRP kinetics in the first days after first surgery and before revision one? To show the time between surgery and revision surgery of these 60 patients can clarify the reason this exclusion criteria.
In the page 4 you describe very well the definition of PJI that you have applied, but it seems that you required a culture positive PJI and you discard patients with inflammatory signs highly suspicious of PJI. Are they in the non-infection group? I would appreciate an explanation because it seems you limit your study only to patients with PJI by highly virulent organisms and this can limit its generalization.
In the other way I haven´t understood how you classified the patients as not infected. Which surveillance system or criteria did you use to define a patient as not infected and how long was their follow-up. How many patients of the 60 excluded by revision surgery criteria were infected and how many were not?
Who applied the definitions of infected vs. not infected? Infection control practitioners? The surgeons of the clinical team? One or more persons of the investigation team? It is very clear that Infected group had PJI but I think further clarification is needed to know how not infected where defined.
In reference to the results preoperative CRP mean and maximal value are higher in infected group as it was the number of days admitted before surgery. Despite it is a not statistical significant difference have the authors any idea why infected patients were admitted a mean 3.38 days (with a high SD) before of the surgery.
I hope you find my questions and comments useful to improve your paper. Best regards
Author Response
Reviewer Comments: Reviewer 2
First of all, I want to congratulate the authors for your work. I think it is very interesting to work on clinical challenges and try to demonstrate that just with CRP kinetics we can anticipate a so difficult diagnosis as it is PJI.
We thank the reviewer for this positive evaluation of our work.
In the abstract you say that you aim to "validate" de value of CRP to predict of PJI. I think what you describe very well are the different approaches to obtain maximal value of CRP kinetics in your series, but to validate the usefulness of your best approach, I think you have to apply in a different cohort of the one that you have used to obtain sensitivity and specificity in your study.
We agree with the reviewer that a confirmation of our approach in a different cohort would increase the clinical applicability. Unfortunately, we utilized all of the available patient data at our level-I endoprosthesis center. We subsequently changed “validate” in the abstract into “determine” and added following sentence to the limitations: “In addition, the results of this study will be tested prospectively in a different patient cohort.”.
In the inclusion and exclusion criteria I haven´t understood why revision surgeries were excluded. The 60 cases were so precocious that didn´t permit to measure the CRP kinetics in the first days after first surgery and before revision one? To show the time between surgery and revision surgery of these 60 patients can clarify the reason this exclusion criteria.
We apologize for the lack of clarity. The issue with the revision surgeries was that most of the respective patients were sent from other clinics with a primary or secondary infection. They were admitted in very heterogeneous conditions, sometimes septic. Patients who had undergone primary pre-operation in external clinics were referred to clinically and chemically proven PJI. However, retrospective access to the externally collected CRP kinetics was not possible due to a lack of digital archiving. Due to the absence of comparability, these patients were therefore excluded. Furthermore, since serum values were not comparable between our laboratory and the laboratory from other clinics, and their mode of implantation was partly different, we did not include these patients and do not have these data. Likewise, we do not have the (reliable) date from the first surgery in another clinic for all of these patients. We regret that we cannot provide these.
In the page 4 you describe very well the definition of PJI that you have applied, but it seems that you required a culture positive PJI and you discard patients with inflammatory signs highly suspicious of PJI. Are they in the non-infection group? I would appreciate an explanation because it seems you limit your study only to patients with PJI by highly virulent organisms and this can limit its generalization.
We thank the reviewer for the opportunity to clarify this important point. Patients with clinical or laboratory signs of an infection first received a sterile joint puncture. In case of a positive result, they were assigned into the “infection” group, and a revision surgery followed. In order to clarify this for the reader, we added following explanation: “Patients were also assigned into the “infection” cohort by a positive joint puncture results as described below.”. In case of steadily rising signs of infection, but a negative joint puncture, a revision surgery was anyhow performed. Obvious signs of an infection (like pus) also led to the classification into the “infection” group. To clarify this, we added “[In order to classify a patient to the “infection” cohort, the detection of an organism on a specimen during revision surgery] or obvious signs of a purulent infection during revision surgery [was necessary.]” to the methods. Fortunately, we always detected a specimen in these cases (Tab. 5). We hope that these statements make our approach easier to understand.
In the other way I haven´t understood how you classified the patients as not infected. Which surveillance system or criteria did you use to define a patient as not infected and how long was their follow-up. How many patients of the 60 excluded by revision surgery criteria were infected and how many were not?
We thank the reviewer for this question. Serum controls were collected regularly, on average every 2.43 days. In case of a steady rising CRP value and/or clinical deterioration, a joint puncture followed. Since we stopped collecting CRP values at the day of discharge, we cannot completely exclude occurring infections after dismissal. However, regarding the setting of our hospital, this seems highly unlikely. If a patient was admitted to the hospital after earlier dismissal and appeared to have an infection, this was classified an “infection”. Of the 60 revision surgeries that were excluded, no one had a primary hip endoprosthesis infection in our hospital. These were either primarily operated in another clinic or revisions which had their primary hip endoprosthesis implantation before 01.01.2016. We feel that the heterogeneity of cases from other hospitals prevents from including these patients into our study. To clarify this, we added following sentence to the methods “Every excluded revision surgery was either primary operated in another clinic or implanted before the beginning of our study (01.01.2016).”.
Who applied the definitions of infected vs. not infected? Infection control practitioners? The surgeons of the clinical team? One or more persons of the investigation team? It is very clear that Infected group had PJI but I think further clarification is needed to know how not infected where defined.
We apologize for the lack of clarity. Indeed, the two first and the last author decided upon the criteria now more extensively elaborated in the methods section. To clarify this, we added “[All patients were divided into a “non-infection” and “infection” (acute early) group] by the authors MPM, IJB and DS.” to the methods.
In reference to the results preoperative CRP mean and maximal value are higher in infected group as it was the number of days admitted before surgery. Despite it is a not statistical significant difference have the authors any idea why infected patients were admitted a mean 3.38 days (with a high SD) before of the surgery.
We thank the reviewer for this question. The authors hypothesize that any preoperative concern (i.e. slightly raised CRP values, like the preoperative CRP [mean] and preoperative CRP [max] would suggest) might have led to intensified preoperative testing before “clearing” for the operation theater. Extensive microbiological, gynecological and neurological councils might have delayed the operative procedure. But since we did not study this aspect in detail, this has to remain speculative.
I hope you find my questions and comments useful to improve your paper. Best regards
We thank the reviewer for the questions and think that with these suggestions, we were able to improve the manuscript substantially and help the reader to better understand our methodology.

Round 2
Reviewer 1 Report
Dear Authors,
the layout of the manuscript was improved and the format was redesigned as suggested.
Additionally, you rewrote your conclusion section and mentioned that the CRP values should not be used as a single parameter to diagnose an PJI and furthermore should not be used for the decision to perform a revision.
Therefore, I would recommend your manuscript for publication!
Author Response
We thank the reviewer for these kind words and the help to improve our study.
Reviewer 2 Report
Thank you for your effort claryfyin
Thank you for your effort clarifying my questions.
The only question is not clear for me yet is which were the criteria to classify a patient as non-infected, probably because of the characteristics of your health system and patient flows between institutions. This can be another weakness of the study and probably you can lose those patients with less virulent microorganisms who present symptoms later. If this is the case, the generalization of the results will be lower.
Author Response
We thank the reviewer for this comment. Reanalyzing our patient cohort, the last patient was dismissed on December 23rd 2020. The final inclusion date was December 31st 2020. Although this data cannot be used for the study, we further observed every patient with a hip endoprosthesis infection until August 31st 2020. None of the patients th dismissed with a primary hip endoprosthesis came into the hospital with an infection, even until then. However, as indicated by the reviewer, there could still be patients lost due to symptoms developing later.
Infections after prosthetic joints are traditionally classified as early (until 3 months after surgery), delayed (3-24 months after surgery) and late (>24 months after surgery) (Zimmerli W, Trampuz A, Ochsner PE. Prosthetic-joint infections. N Engl J Med. 2004), although this classification has also raised to some critique (Osmon DR et al.; Infectious Diseases Society of America. Executive summary: diagnosis and management of prosthetic joint infection: clinical practice guidelines by the Infectious Diseases Society of America. Clin Infect Dis. 2013).
Due to the prominent position within the region (and the DRG-related economic impact of a PJI), and the (forced) documentation of these cases within the EPRD (Endoprothesenregister Deutschland), it is highly unlikely that we missed an acute infection, and improbably to miss delayed and some late infections. But the reviewer is right. We might have missed patients with a late infection.
Subsequently, we added following sentence to the limitations: “Nevertheless, some less virulent microorganisms may have led to the delayed development of symptoms and a delayed or late infection may have been missed, lowering the generalization of our results.”.
In addition, for the highlights, we added “[On day 11 and 14 post operation, the CRP values were higher in the group with an] early [infection.]” and “[As a single parameter, maximum CRP predicts an] early [infection with 75% sensitivity.] and “[A multinominal logistic regression can predict an] early [infection with a sensitivity and specificity of 87.50% and 78.85%, respectively.]”
In the abstract, we added “[A multinominal logistic regression, however, was able to predict an] early [infection with a sensitivity and specificity of 87.5% and 78.9%.]” and [Including five parameters, an] early [infection can be predicted on day 5 after the operative procedure with 87.5% sensitivity, while it can be excluded with 78.9% specificity].
In the conclusions, we added “[An] acute [infection can be accurately predicted with a binary logistic regression resulting in a sensitivity of 75% and a specificity of 80%.]”. and “[Summarizing, a one-phase exponential decay can predict CRP kinetics, and an] acute [infectious complication can be reliably projected by using just five parameters.]”
We thank the reviewer for pointing out this aspect and hope that we could improve the submission substantially by adding this limitation and clarifying that our approach is applicable for the acute periprosthetic joint infection.